# Clinical and Pathological Manifestation of the Oral Mucosa in Institutionalized Children from Romania

**DOI:** 10.3390/ijerph192315564

**Published:** 2022-11-23

**Authors:** Alexandra Mihaela Stoica, Csinszka Andrea Kovacs-Ivacson, Oana Elena Stoica, Liana Beresescu, Monica Monea

**Affiliations:** 1Department of Odontology and Oral Pathology, George Emil Palade University of Medicine, Pharmacy, Science, and Technology of Târgu Mures, 540139 Târgu Mures, Romania; 2Department of Pedodontics, George Emil Palade University of Medicine, Pharmacy, Science, and Technology of Târgu Mures, 540139 Târgu Mures, Romania; 3Department of Preventive and Community Dentistry, George Emil Palade University of Medicine, Pharmacy, Science, and Technology of Târgu Mures, 540139 Târgu Mures, Romania

**Keywords:** morsicatio buccarul, oral pathology, cheek biting, nervous tics, plaque-induced gingivitis, geographic tongue

## Abstract

Institutionalized children, regardless of their age, are prone to psychiatric disorders, compared to those who live in two-parent families, due to the unfavorable physical and psychological conditions in which they develop. Unpropitious psychological conditions affect the child’s general health and induce local manifestations that can be found in the oral cavity, affecting both soft tissues and teeth. Therefore, the purpose of our study was to assess and diagnose a series of pathological manifestations of the oral mucosa associated with poor living conditions or closely related to psychological stress. The clinical examination was performed by 4 specialist dentists, who consulted 150 children representing the study group and 52 children representing the control group, all having ages between 14 and 18 years old and meeting the same inclusion criteria. From the control group, 62.5% presented plaque-induced gingivitis (26.6% girls, 35.8% boys), 26.6% presented mucosal lesions produced by lip and cheek biting (23.3% girls, 35.8% boys), and 11.7% presented geographic tongue (6.6% girls, 5% boys), herpes simplex eruptions (3.3% girls, 4.2% boys) and oral ulcerations (12.5% girls, 10.8% boys). Morsicatio buccarul is a chronic, self-mutilating habit, currently becoming one of the most common tics encountered among institutionalized children. Furthermore, these children do not benefit from constant support and guidance to perform their dental hygiene, so the incidence of gingivitis induced by bacterial plaque and calculus is very high.

## 1. Introduction

One of the fundamental rights of any person is the general feeling of happiness and satisfaction in life and their environment, including aspects such as health, free time, culture, rights, values and beliefs, aspirations and basic conditions of life [1]. From the health point of view, quality of life reflects the feeling of physical and mental health, but also the ability to react to factors in physical and social environments, presenting a great degree of subjectivity; therefore, it can be difficult to measure [2]. 

Quality-of-life measurement tools related to oral health are difficult to achieve for children and adolescents because many methodological and conceptual problems arise. For example, understanding a child regarding their health and illness depends on their age and their social, linguistic, emotional and cognitive development. Also, children become more and more aware with age, regarding the psychosocial impact of changes in the facial area and dentition. To measure and compare these changes in different developmental stages and at different ages can be difficult [3,4].

Psychological stress affects a person’s physiological functioning to a significantly dangerous point. An important proportion of the population suffers from psychological disorders such as anxiety and depression, having a negative influence on the physical health of the individual [5,6]. Stress is defined as an emotional, physical or mental response to an event that causes physical or mental tension [7]. A series of chemical reactions take place during stress, and these allow the various organs to withstand the irritating stimuli. Prolonged exposure to these biochemical changes predisposes the body to disorders [8]. A psychosomatic condition involves both the mind and the body as they constantly influence each other.

Diseases of the oral cavity with psychosomatic etiology have been known in medicine for a long time, with the mouth having an important relationship with human instincts and, therefore, with unconscious brain activity. Emotions can represent risk factors in the initiation or progression of oral pathologies [9].

Although the state of Romania has made significant progress in reducing the number of abandoned children over the last three decades, the country still has one of the largest case numbers in the region. At present, the state takes care of the needs of over 56,000 children. Out of these, approximately 18,000 are in foster care, 13,900 are in the care of relatives, 4800 are in the care of other families and 18,500 are in state institutions [10]. About 3000 of them are adoptable. It is worth noting that most of the children in state care are so-called “social orphans”, because the mother is alive and known in the case of more than 90% of the children, and 48% of the children also have fathers. Vulnerable families struggling to keep their children and avoid separation from them need support. Approximately 5000 children enter the care system each year [11]. 

We consider that abandoned children are in disadvantage from the start in terms of quality of life because they have to go through the shock of abandonment at a young age. In many cases, caregivers or adoptive parents make huge efforts to help the little ones to overcome these psychological thresholds, and some are partially healed. Often, a good material life and an above-average socio-economic status cannot really compensate for the traumas accumulated in early childhood, and these traumas latently affect the general health of young people in various forms or materialize in different tics or self-mutilating behaviors like tongue, lip and cheek biting, or various lesions of the oral mucosa [12]. 

One of the symptoms of a difficult life period is repetitive harmful behavior focused on the body, comparable to hair pulling, nail biting and frequent blinking. Chronic biting of the checks (buccal mucosa), medically known under the term chronic keratosis or morsicatio buccarul, is classified in the Diagnostic and Statistical Manual of Mental Disorders (DSM-5) under obsessive–compulsive and related disorders and is complementary to the types of problems related to anxiety. 

Repetitive body-focused behaviors, such as cheek biting, most often begin in early childhood and can last throughout adulthood. Since we often cannot escape the stress of everyday life and the resulting anxiety associated with it, behaviors such as this manifest as a subconscious solution to ease emotional overload [13]. 

Another essential aspect that is discussed intensively and is well studied is the dental status and dental health of these young people. It is known that tooth decay is a frequent chronic disease of children and young people, especially in those associated with poor living conditions or living in limited socio-economic conditions. The incidence of caries in children increases even more in less-developed countries whose citizens do not benefit from a system of dental training and prevention. 

These cases can be supported and helped by creating a system and a medical guide for dentists, who, knowing the risk factors, try to reduce the incidence of caries and emphasize oral prevention, especially in children. The development of specific prevention protocols for certain age groups or adapted to the socio-economic situation and especially supporting children without parents could considerably reduce the incidence of both dental caries and gingivitis induced by bacterial plaque [14].

The oral cavity, together with the tongue, soft tissues and teeth, is an indicator of the status of the systemic function of the whole body, because at this level, the symptoms of about 300 diseases can manifest or appear [15]. Currently, the quality of life for patients of all ages is recognized as an important component of oral and general health, and the impact of various daily stress factors can influence the integrity of the oral soft tissue anatomy. In addition to psychological and psychosocial factors, there is a long list of factors that locally affect the mucosa of the oral cavity, such as poor hygiene, inadequate nutrition, and alcohol and tobacco consumption [16].

Most oral diseases can be prevented by a reduction in the level of exposure to major risk factors and promoting a healthy lifestyle. Socio-behavioral factors and environment have a significant role in oral diseases, but these can be modified by programs promoting healthy behavior and maintaining a state of proper oral hygiene. 

Also, by increasing the awareness of the young population of the importance of oral health and factors of risk, and by giving them the knowledge necessary to maintain oral health, we can reduce the incidence of oral pathology among children and adolescents [17].

Therefore, the purpose of our study was to assess and diagnose a series of pathological manifestations of the oral mucosa associated with poor living conditions or closely related to psychological stress and to support oral and dental prevention in these disadvantaged cases.

## 2. Materials and Methods

### 2.1. Study Design

This study was carried out during a period of 2 years and 3 months, from February 2020 until May 2022, in the Private Dental Clinic StoicaMed in Targu-Mures, Mures County, Romania. The design of the study and methodology were approved by the clinic ethics committee (no. 40/03. Feb. 2020). The written consent of the legal guardians of the institutionalized children was obtained. In order to obtain valid results, we also created a control group of noninstitutionalized children from various families (urban and rural zones) all studying at the local high school in Targu-Mures. In order to start the study, we also obtained the written consent of all children’s parents included in the control group. For the study group, we selected 156 children, both boys and girls in equal number, meeting the following inclusion criteria: ages between 14 and 18 years old, clinically healthy, cooperative (willing to work together or cooperate by answering to medical questions) and having been in the orphanage for at least 5 years. For the control group, we selected 60 children, boys and girls in equal number, with ages between 14 and 18 years old, also cooperative and clinically healthy. For all children, the exclusion criteria were age under 14 or older than 18 years, having recently been institutionalized (for the study group), or the presence of the following general and systemic issues: associated psychiatric and mental disorders, genetic or congenital syndromes, uncooperative behavior or receiving any type of medicamentous treatment (Figure 1).

### 2.2. Clinical Examination and Assessment

The clinical examination of the oral mucosa and data recording was performed by 4 specialist dentists with more than 5 years of experience in Oral Pathology and Pedodontics. The clinical oral examinations were performed using disposable dental instruments (mirrors) and proper medical equipment (ffp2 masks, gloves, medical caps). For each young patient, the inspection of the oral mucosa respected the same order of assessment, starting with the mobile anatomical parts (labial mucosa, buccal mucosa, floor of the mouth, tongue, soft palate) and continuing with the fixed oral tissues (hard palate and gingival tissue). Each clinical modification of the physiology of the oral mucosa was assessed in detail and photographed. For every young patient presenting a pathological mucosal lesion, we assessed the time of onset and recurrency of having an oral lesion, the presence of pain while eating or speaking, and the connection between a stressful period and the presence of the oral lesion. 

### 2.3. Statistical Data Analysis 

Jamovi statistics software (2.2.5.0 3rd generation, Sydney, Australia) for Windows 2020 was used, including independent *t*-tests for the descriptive analysis of all data obtained. For the significance level, the value of 0.05 was chosen, and p was thus considered significant if *p* ≤ 0.05. 

## 3. Results

The study group initially included 156 children, of which 36 were excluded from the study for not meeting our inclusion criteria (8 children were recently institutionalized, 14 were younger than 14 years of age, 3 children were diagnosed with genetic disorders at birth, and 11 children did not cooperate and could not be examined). 

The control group initially included 60 young patients, from which were excluded 2 children who did not agree be examined, 3 children diagnosed with congenital disorders and another 3 whose parents did not agree to sign the written consent, leaving 52 eligible children included in the study. The study group included 54 (45%) boys and 66 (55%) girls the control group included 25 (48.1%) boys and 27 (51.9%) girls.

The number of young people who ended up being included in the study (sample size) started from collecting the maximum number of cases to which we had access. Collaboration with child shelter institutions is not easy, and despite our efforts, the number of institutionalized children who met the inclusion criteria reached a maximum of 120. In a similar way, we proceeded to gather as many young people as possible to form the control group, selecting children from two-parent families who also complied with the same inclusion criteria established initially.

The following Table 1 presents the division of all eligible children into groups according to gender for both the study and control groups. The division according to age of all patients included in the study for both the study and control groups is represented in the following Table 2. 

After carrying out 120 examinations and inspections (for the patients included in the study group) of the oral mucosa, we discovered the following pathological aspect changes: lesions of the mobile mucosa and/or lips (Figure 2) due to conscious biting either by the development of this nervous tic or chronic, agitated and nervous behaviors, plaque-induced gingivitis (Figure 3), oral ulceration (Figure 4), herpes simplex lesions (Figure 5) and geographic tongue (Figure 6) all presented in Table 3. For the 52 examinations representing the control group, the oral mucosa pathologies included geographic tongue and plaque-induced gingivitis (Table 4). 

Out of all the young people included in the study group, 28 girls (23.3%) and 4 (3.3%) boys presented chronic keratosis (Morsicatio buccarul) on examination, representing 26.6% of the entire study group. Out of these total 32 cases, 10 (31.2%) declared that biting their cheek is unintentional and they do not do it consciously, and 22 adolescents (68.7%) said that cheek and lip biting creates a pleasant feeling and they do it both consciously and unconsciously. Out of all 32 cases of chronic keratosis, for 5 (15.6%) persons the lesions were present on the mucosa of the inferior lip only, and the remaining 27 (84.4% with *p* < 0.01, a significant difference) prefer biting the mucosa of the inferior lip as well as the mobile mucosa of both checks.

The results obtained after interviewing the young patients regarding the sensitivity or discomfort of the lesions present in the oral cavity during speech or mastication in both groups included in the study are presented in the following Table 5.

## 4. Discussion

All the young people who participated in the present study were of ages between 14 and 18 years old, an extremely sensitive adolescent age, with many changes in terms of both physical and mental or intellectual development. In this age group, an essential role is played by the hormonal status, which can induce characteristic changes in the tissues of the oral cavity, mainly on the mobile mucosa, but also on the gingival tissue.

This age group is also essential in building certain behaviors, and specific psychological problems of some adolescents can also be highlighted that have roots in early childhood [18,19].

The study started from the premise that abandoned and institutionalized children go through certain moments of psychological stress, with the fear of abandonment consciously or unconsciously affecting their lives. Often, psychological distress in young children has an impact on their physical health, including the health of the tissues of the oral cavity [20]. The quality of life of children supported by parents who have a comfortable social rhythm and do not live in fear of being abandoned or alone is much higher, having the comfort of a family and a home. Thus, we created a study group including institutionalized children (120 adolescents, boys (45%) and girls (55%)) and a control group represented by children (52 adolescents, boys (48.1%) and girls (51.9%)) from families with both parents present, studying at the same educational institution.

Children and young people under 18 who are institutionalized in Romanian orphanages are not a priority from the point of view of oral hygiene education, and there is no well-organized national system that can support the oral health of these young people. Thus, there are associations that voluntarily help and participate in their development and education, but the volunteers are overwhelmed and exceeded by the large number of abandoned children. These adolescents are not aware and do not invest time and attention into prevention or oral hygiene, which is then deficient—a fact supported by the results of our study. Therefore, gingivitis induced by bacterial plaque, as a pathology of the oral soft tissues, was found in high numbers, in 32 girls (26.6%) and 43 boys (35.8%), with a value of *p* < 0.01 (statistically significant). Plaque-induced gingivitis in the control group was found in 10 girls (19.2%) and 16 boys (30.7%), at lower rates compared to the study group.

According to the results and studies in the literature, boys have poorer dental hygiene than girls; not placing as much interest in the aesthetics of the smile, boys in both the study and control groups had higher incidence of plaque-induced gingivitis.

It is well demonstrated by multiple studies that the oral manifestations of infection with the herpes virus appear when the human body is subjected to the stress of adapting to large social, emotional or just physical differences, a fact that can affect the individual’s life balance and immunity [21,22]. Not by chance, exactly on the background of low immunity, we can have the unpleasant surprise of noticing the appearance of herpes on the lips, especially in children and young people [23,24]. Also, common causes that lead to herpetic relapses include a decrease in immune system effectiveness due to a serious illness, physical or emotional overwork, sleep deprivation, or multifactorial and continuous psychological stress. Of those we examined, only 4 girls (3.3%) and 5 boys (4.2%) presented with oral herpes simplex lesions in the study group. 

Numerous recent publications also studied other manifestations and oral pathologies that include immune factors and stress in their etiopathology [25,26]. Geographic tongue (benign migratory glossitis), which also belongs to this category, is, in general, a harmless condition characterized by the appearance on the surface of the tongue (dorsal face or edges) of areas with an irregular outline, red in color and surrounded by a whitish border [27,28]. The cause of this condition is unknown, but factors such as stress, various hormonal changes or nutritional deficiencies (zinc, vitamin B) can be blamed. Although harmless, benign migratory glossitis can cause pain or burning sensations of the tongue during the consumption of certain foods, such as citrus fruits, spicy foods, strawberries, tomatoes or pineapple, or during alcohol consumption [29]. The presence of geographic tongue was observed in the control group adolescents in 14 girls (26.9%) and 9 boys (17.3%), with girls presenting a higher rate compared to boys.

The study of quality of life among children and adolescents plays an important role in oral and dental health [30,31]. Different authors and studies have concluded that the appreciation of quality of life from a health point of view allows the correlation of parameters indicating “how much” and “how well” an individual lives [32,33,34]. Diseases in the oral-maxillo-facial sphere can affect these parameters, leading to alterations in the perception of self-image, self-esteem and well-being. The social level of each patient and the doctor’s understanding of their needs can, in some cases, influence the decision regarding the choice of treatment [35].

Stressed individuals may develop vicious habits such as biting the lip or cheeks, chronic keratosis (Morsicatio buccarul), or biting even objects that can cause dental attrition, vicious habitual occlusion, bruxism or facial muscle pain. In our study, there was a high percentage of adolescents with chronic keratosis of the cheek, which is a white lesion that can be confused with leukoplakia or vice versa. Therefore, in patients with known risk factors for developing oral premalignant lesions (smoking, poor oral hygiene, alcohol consumption), adjuvant methods such as vital dyes or exfoliative cytology may be used for a final diagnosis [36,37]. There may also be gastric disturbances leading to dental erosion due to vomiting episodes [38,39,40]. Anecdotal reports indicate an increase in the number of young adults suffering from stress-induced oral pathologies [41]. These include xerostomia, lichen planus, bruxism and glossodynia. An abnormal reduction in saliva production leads to xerostomia or dry mouth and is frequently encountered in patients with psychiatric problems. Aphthous stomatitis is another example of a lesion that occurs in stressful situations such as school exams or even during dental treatment [42,43]. Psychiatric disorders and sleep disorders are etiological factors for bruxism. Patients with myofascial pain syndrome report symptoms such as frustration, anxiety, depression and maladaptive behaviors such as pain, inadequate sleep and incorrect eating habits. A burning sensation in the oral cavity is a condition found in patients with psychological, behavioral or sleep disorders [44]. 

The results of our study also support these scientific findings, with chronic keratosis (Morsicatio buccarul) due to lip and cheek biting affecting a significant (*p* < 0.01) number of individuals in the study group (28 girls (23.3%) and 4 boys (3.3%)). This statistic shows that girls present a higher incidence of lesions induced by biting. They also declared that, although they were informed that it is not a healthy tic, cheek and lip biting is most of the time done consciously, to create a pleasant feeling of relaxation when having a depressed or anxious moment. For them, it is an already usual behavior that brings mental comfort, although often these injuries give them discomfort when eating hot or acidic foods [45]. 

After we recorded and examined the pathology of the mucosa, both in the children in the study group and in the children in the control group, we asked each of them if they feel pain or sensitivity during mastication or while speaking and if it makes their daily life uncomfortable. After centralizing the results, we found that the lesion that creates discomfort both in mastication and speaking is oral ulceration, especially if they consume acidic or spicy foods; they consider the feeling of discomfort as a real pain, both in the case of girls and in the case of boys. Also related to the consumption of spicy foods, tongue sensitivity was mentioned in the case of those who presented geographic tongue (10.6% of girls and 9.2% of boys) and in the control group (29.6% of girls and 28% of boys). Related to plaque-induced gingivitis, the sensation described by the young participants in our study was of reduced discomfort or slight sensitivity only in the case of tooth brushing, without painful sensations when chewing or speaking. In total, 13.6% of girls and 1.9% of boys from the control group who presented chronic keratosis mentioned that normally this tic momentarily creates a feeling of relaxation, but after a fresh injury, sometimes much too deep in the tissue, they feel pain both when speaking and during mastication, a fact that does not cause them to stop this vicious habit.

Predominantly, nervous tics appear in childhood, and the family environment is the most important triggering factor [46]. If children are abandoned and live without the support and love of their parents or live in an environment where there are tensions, arguments and contradictory discussions, the child emotionally takes all these situations and reflects them in these tics, as a defense mechanism [47]. If they are treated in time and psychologically supported, in the case of the majority of children, the tics diminish or disappear completely in adolescence; otherwise, the biting behavior of the lips or the mucous membrane of the cheeks persists and worsens with age, possibly degrading into pathological sessions with malignancy potential. Treatment for patients with oral or extraoral self-injurious habits depends on many factors, such as their age, severity, old habits and cooperative behavior. It is important that self-harm is taken seriously and that the help and support provided is appropriate. As future objectives, we consider that correct oral hygiene in patients from all types of life styles and socio-economic situations is essential to prevent possible oral complications. The use of an electric toothbrush, gels and mouthwashes with an antimicrobial character, and also improving the quality of food, can reduce the plaque deposits and prevent bacterial gingivitis and further complications. In the case of institutionalized children, this help must come from the caregivers or volunteers who are visiting the orphanages and offer their time to educate and teach the children the importance of oral hygiene [48].

## 5. Conclusions

Abandoned children who enter foster care centers have special needs in all areas of health, of which only a small part is provided, so that only a minority receives the psychological and physical support necessary to prevent the aggravation of their conditions.

Institutionalized children are not supported and helped to perform their dental hygiene, so the incidence of gingivitis induced by bacterial plaque and calculus is very high among these children, with 62.4% of all such children presenting poor dental hygiene.

Psychiatric disorders have intraoral injuries as side effects, or oral pathologies like geographic tongue (11.6%), oral ulcerations (23.3%) or chronic keratosis, some of which are self-mutilating and others of which appear through the general disturbances caused by stress in the body. The incidence of lesions due to morsicatio bucarum and labiorum in our study was 26.6%, with a higher rate among girls; this is carried out unconsciously or compulsively as a nervous tic, offering a pleasant and relaxing feeling, and is currently becoming one of the most common tics encountered among institutionalized or non-institutionalized children. 

Particular attention must be paid to vulnerable population categories by offering preventive services, especially to people living in poverty. Improvements could come with the implementation of protocols and education sessions for oral health and prevention with the help and collaboration of private providers.

## Figures and Tables

**Figure 1 ijerph-19-15564-f001:**
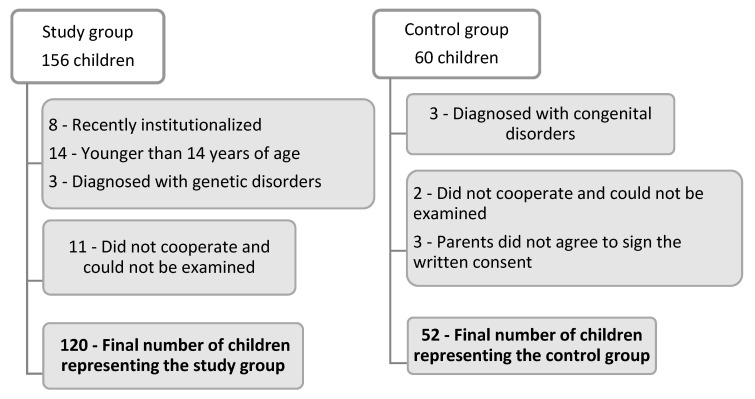
Diagram illustrating the selection of the young patients included in the study.

**Figure 2 ijerph-19-15564-f002:**
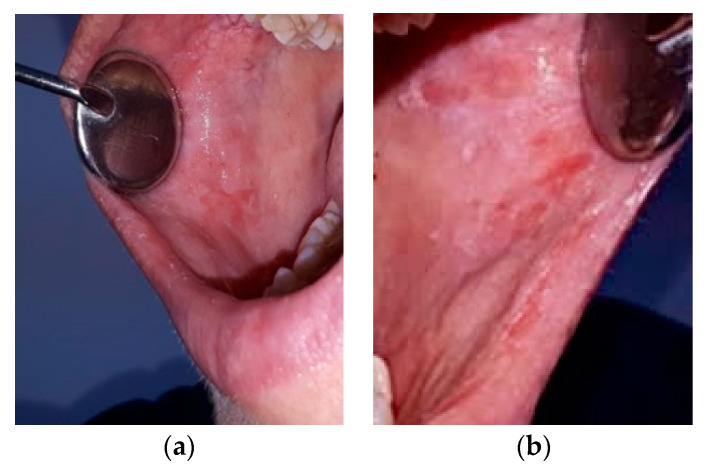
Chronic keratosis (Morsicatio buccarul) in a 17-year-old girl: (**a**) keratotic lesion on the oral mucosa of the right cheek; (**b**) keratotic lesion on the oral mucosa of the left cheek.

**Figure 3 ijerph-19-15564-f003:**
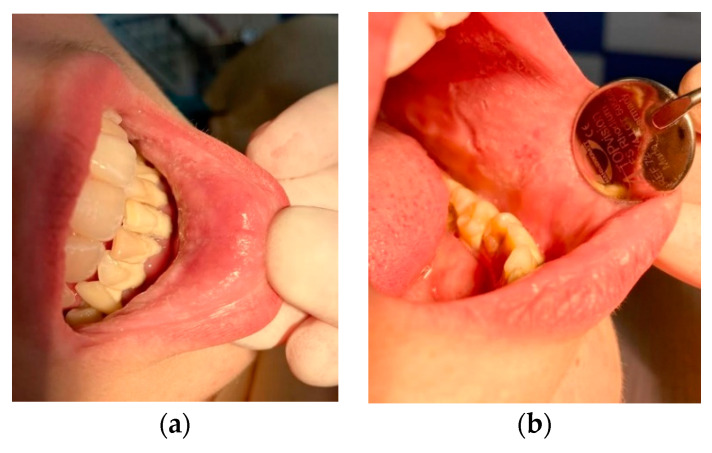
Chronic keratosis (Morsicatio buccarul) in a 16-year-old girl: (**a**) keratotic lesion on the oral mucosa of the inferior lip/plaque-induced gingivitis; (**b**) keratotic lesion on the oral mucosa of the left cheek.

**Figure 4 ijerph-19-15564-f004:**
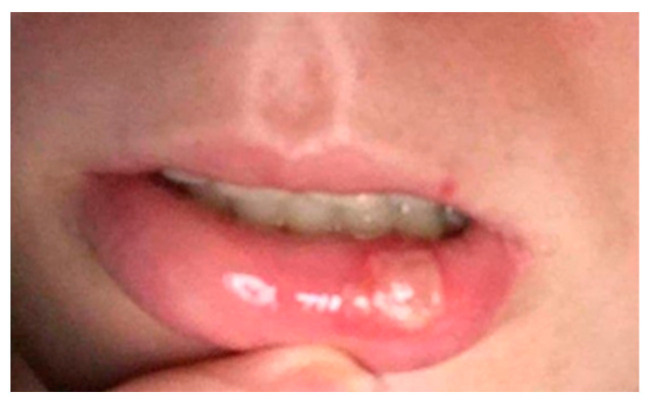
Oral mucosa ulceration on the inferior lip affecting a 16-year-old girl.

**Figure 5 ijerph-19-15564-f005:**
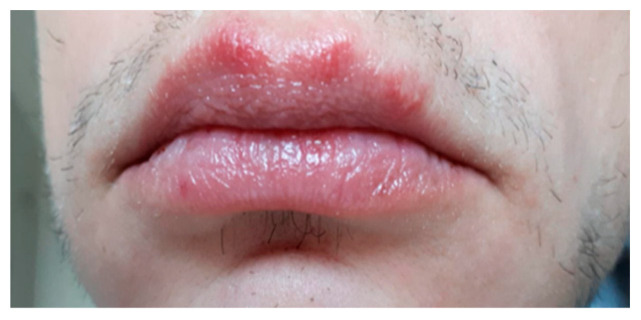
Herpes simplex lesions on the upper lip affecting a 17-year-old boy.

**Figure 6 ijerph-19-15564-f006:**
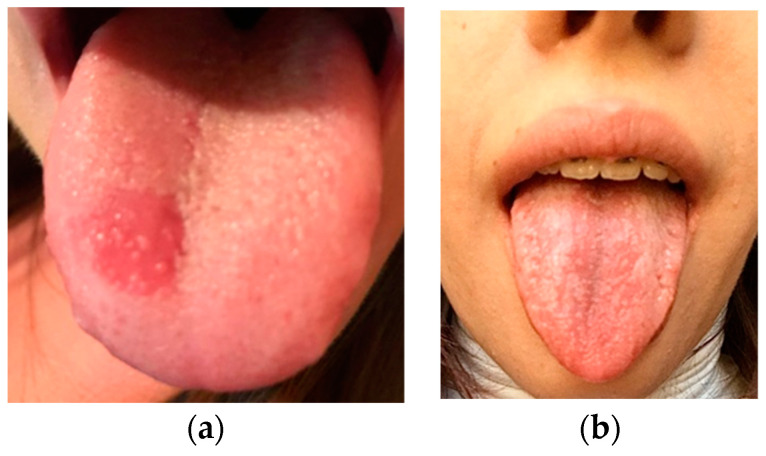
(**a**) Geographic tongue affecting a 16-year-old girl; (**b**) Geographic tongue affecting a 17-year-old girl.

**Table 1 ijerph-19-15564-t001:** Division of all eligible children according to group and gender.

Gender	Study Group	Control Group
Boys	54 (45%)	25 (48.1%)
Girls	66 (55%)	27 (51.9%)

**Table 2 ijerph-19-15564-t002:** Division according to age of both study and control groups.

Age	Study Group Total 120	Control Group Total 52
**14 years old**	33 (27.5%)	12 (23.1%)
**15 years old**	27 (22.5%)	17 (32.7%)
**16 years old**	40 (33.3%)	13 (25%)
**17 years old**	20 (16.6%)	10 (19.1%)

**Table 3 ijerph-19-15564-t003:** Oral mucosa pathologies present in patients representing the study group.

Oral Mucosa Pathology	Study Group		Total
	Girls	Boys	
Geographic tongue	8 (6.6%)	6 (5%)	11.6%
Herpes simplex lesions	4 (3.3%)	5 (4.2%)	7.5%
Oral ulceration	15 (12.5%)	13 (10.8%)	23.3%
Plaque-induced gingivitis	32 (26.6%)	43 (35.8%)	62.4%
Chronic keratosis (Morsicatio buccarul) significant difference *p* < 0.01	28 (23.3%)	4 (3.3%)	26.6%

**Table 4 ijerph-19-15564-t004:** Oral mucosa pathologies present in persons representing the control group.

Oral Mucosa Pathology	Control Group		Total %
	Girls	Boys	
Geographic tongue	14 (26.9%)	9 (17.3%)	44.2%
Plaque-induced gingivitis	10 (19.2%)	16 (30.8%)	50%

**Table 5 ijerph-19-15564-t005:** Oral mucosa pathology pain or sensibility in patients representing the study and control groups.

Oral Mucosa Pathology	Study Group Pain/Sensibility
Geographic tongue	Present 7 girls (10.6%) 5 boys (9.2%)
Herpes simplex lesions	Present 1 girl (1.5%) 2 boys (3.7%)
Oral ulceration	Present 2 girls (3%) 3 boys (5.5%)
Plaque-induced gingivitis	Present 16 girls (24.24%) 19 boys (35.2%)
Chronic keratosis (Morsicatio buccarul)	Present 9 girls (13.6%) 1 boy (1.9%)
Oral mucosa pathology	Control Group Pain/Sensibility
Geographic tongue	Present 8 girls (29.6%) 7 boys (28%)
Plaque-induced gingivitis	Present 5 girls (18.5%) 11 boys (44%)

## Data Availability

All data regarding this manuscript can be checked with the corresponding authors.

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
