# Peer review of "Clinical and Pathological Manifestation of the Oral Mucosa in Institutionalized Children from Romania"

_ijerph, 2022, doi:10.3390/ijerph192315564_

Round 1

Reviewer 1 Report

Thank you for the opportunity to review this interesting study on the clinical and pathological manifestation of the oral mucosa in institutionalized children from Romania. This study includes a vulnerable population of children, and is an important area of research. Some clarity in the reporting of results would be beneficial. It would be fascinating to see a table with the number of lesions/by gender/at a specific time and event period of their lives. 

Abstract

-      Please change ‘specializes’ to specialist 

-      For consistency, maybe 11.7% (instead of 11.66%) as the other percentages are to one decimal point

-      Delete ‘constantly’ in this sentence ‘…these children do no benefit of constantly support…’

Introduction

-      Please include references for these paragraphs “In present it takes care of the needs of over 56,000 children. 60 Out of these, approximately 18,000 are in foster care, 13,900 in the care of relatives, 4,800 61 in the care of other families and 18,500 in state institutions”.

-      Please include reference here “…about 300 diseases can manifest or appear”.

-      Please change ‘de’ to ‘the’

Method

-      Include the country in the sentence after ‘…Mures County, Romania’

-      What does ‘cooperative’ mean in the inclusion criteria? Add a definition of what this is please.

-      Figure 1: Please edit ‘din’ to ‘did’

-      What type of statistical analysis was performed? Chi-square ? 

Results

-      Where is the table/findings or data for this reported in the method ‘… time of onset and recurrency of having an oral lesion, the presence of pain while eating or speaking, the connection between a stressful period …’. How many children had presence of pain/what types and how many stressful events were recorded?

-      Any demographic information about dental hygiene, preventive behaviours? 

-      Edit ‘Aldo’ to ‘Although’

Discussion

-      Grammatical - please edit the sentence ‘their life’s’ . 

-      Please clarify this sentence ‘At level personally, influences…..’ This is unclear. 

Conclusion

-      Where is this finding in the results section for this: ‘62.5 % of the presenting poor dental hygiene’. Is there a table/data?

Author Response

Dear Reviewer,

        We appreciate your interest in our manuscript and we want to thank you for the guidelines offered for improving it. We attache our response.

Thank you so much!

Reviewer 2 Report

Manuscript of considerable interest for the dental sector. Before proceeding with the evaluation for a possible publication, a major revision is required.

Abstract, highlight statistically significant data more

Few keywords, to update specifics

Introduction: the section concerning the contributing causes with lesions of hard tissues is missing, both on the deciduous elements and on the permantents, as already studied by the research group of Prof. Scribante

Materials and Methods: how was the sample size calculated?

Results: very confusing, reorganize them so that the common reader can interpret them, highlighting the statistically significant data

Discussion, to add as future objectives, the use of postbiotic-based gels, toothpastes based on tindalised probiotics, ozonated water, etc., as already studied by the research group of Prof. Butera et al.

Conclusion reforum them by adding proactive action

Bibliography, add required references

Author Response

(The authors gave the same response as above.)

Reviewer 3 Report

Dear authors!

The issue of the presented study is very valuable and important for the future optimization both for the medical and social spheres. However, manuscripts’ methodological part requires improvements.

Comments are in the attached file.

Author Response

(The authors gave the same response as above.)

Round 2

Reviewer 2 Report

The manuscript has been correctly revised, it can be published